# Correlation between the Control of Blood Glucose Level and HbA1C and the Incidence of Surgical Site Infection after Emergent Surgery for the Lower Limb Fracture among Type II DM Patients Aged More Than 50 Years Old

**DOI:** 10.3390/jcm11195552

**Published:** 2022-09-22

**Authors:** Wei-Hung Wang, Tsung-Cheng Hsieh, Wen-Tien Wu, Ru-Ping Lee, Jen-Hung Wang, Kuang-Ting Yeh

**Affiliations:** 1School of Medicine, Tzu Chi University, Hualien 970374, Taiwan; 2Institute of Medical Sciences, Tzu Chi University, Hualien 970374, Taiwan; 3Department of Orthopedics, Hualien Tzu Chi Hospital, Buddhist Tzu Chi Medical Foundation, Hualien 970473, Taiwan; 4Department of Medical Research, Hualien Tzu Chi Hospital, Buddhist Tzu Chi Medical Foundation, Hualien 970473, Taiwan; 5Graduate Institute of Clinical Pharmacy, Tzu Chi University, Hualien 970374, Taiwan

**Keywords:** type II DM, perioperative control of HbA1c and AC glucose, receiver-operating characteristic (ROC) curve, emergent orthopedic surgery

## Abstract

This is the first study focusing on perioperative blood glycemic monitoring for the incidence of surgical site infection (SSI) among patients with type II DM (T2DM) during the 1-year follow-up after emergent orthopedic surgery. We retrospectively collected the data of 604 patients who had received surgery for unilateral lower limb traumatic fracture from January 2011 to January 2021, including 215 men and 389 women with a mean age of 71.21 and a mean BMI of 25.26. In total, 84 (13.9%) of them developed SSI during the 1-year follow-up. Higher preoperative and postoperative -3-month hemoglobin A1c (HbA1c) and AC blood glucose and the presence of rheumatoid arthritis were all associated with increased rates of SSI. The thresholds for predicting SSI were the following: (1) preoperative HbA1c > 7.850% (area under curve [AUC] = 0.793); (2) postoperative HbA1c > 6.650% (AUC = 0.648); (3) preoperative AC blood glucose > 130.50 mg/dL (AUC = 0.773); and (4) postoperative AC blood glucose > 148.5 mg/dL (AUC = 0.709) by receiver-operating characteristic curve method. These findings may provide a useful control guideline for patients with T2DM older than 50 years old and who received surgery for a lower limb fracture in the prevention of postoperative SSI.

## 1. Introduction

The international diabetes federation reports that approximately 536.6 million (about 10.5% of the world population) adults had diabetes mellitus (DM) worldwide in 2021; this figure is expected to rise to 783.2 million (12.2%) by 2045 [1]. Systematic reviews and meta-analyses have indicated that inadequate glycemic control is associated with a higher risk of postoperative complications and worse outcomes in various surgical specialties [2,3,4]. Surgical site infection is a highly severe orthopedic complication that leads to poorer patient outcomes, and it is more prevalent in patients with type II DM (T2DM) receiving orthopedic surgery [5,6,7]. However, research on the optimal threshold for hemoglobin A1c (HbA1c) remains inconsistent [4,8,9], and no consensus has been reached regarding guidelines for a recommended HbA1c value at which surgeries should be put off [10,11,12]. Likewise, perioperative blood glucose targets vary between guidelines [13,14,15]. Despite numerous studies establishing the effect of hyperglycemia on postoperative outcomes, most of them have focused on determining the ideal preoperative HbA1c level and blood glucose [16,17,18,19] for a low complication rate in elective surgery. Postoperative 3-month glycemic status, which can be a vital indicator of DM control quality, should be necessary for evaluating surgical outcomes of patients with T2DM but is seldom discussed in the literature. Furthermore, even studies that have investigated preoperative and postoperative glycemic statuses together have primarily only focused on short-term perioperative blood glucose [20,21]. Postoperative long-term glycemic markers such as HbA1c have been seldom investigated. An ideal operation time under ideal DM control status may be difficult to achieve for patients with T2DM who have traumatic fractures and require emergency surgery. However, the quality of postoperative DM control may be even more critical for reducing surgical complications than for those who receive elective or scheduled surgery. Therefore, we aim to evaluate whether blood glucose and HbA1c levels obtained both preoperatively and 3-months-postoperatively are associated with the incidence of surgical site infection (SSI) during postoperative 1-year follow-up and evaluate the performance of HbA1C or blood AC glucose in the prediction of SSI. Our findings may provide valuable information for postoperative glucose management in T2DM orthopedic trauma patients and in perioperative glycemic evaluation and management of patients with T2DM receiving elective orthopedic surgery.

## 2. Materials and Methods

The ethics committee of the Institutional Review Board of the Buddhist Tzu Chi Medical Foundation Hualien Tzu Chi Hospital approved the study’s retrospective protocol (IRB 111-016-B). The study protocol was implemented in accordance with the Declaration of Helsinki. We conducted a retrospective, observational single-center study from January 2011 to January 2021. Patients were included if they (1) were aged 50 years or older, (2) had T2DM, (3) had received surgery for unilateral lower limb traumatic fracture within 2 weeks of trauma, (3) had available data of their preoperative blood hemoglobin A1c (HbA1c) and AC glucose levels, and (4) had available data of their postoperative 3 months blood hemoglobin A1c (HbA1c) and AC glucose levels. Data on the patients’ demographic characteristics, including age, sex, BMI, and medical comorbidities, were also collected. Data on medical comorbidities, including hypertension, dyslipidemia, coronary artery disease (CAD), cerebral vascular accident, hepatic disease, chronic renal failure (CRF), and rheumatoid arthritis (RA), were analyzed. We excluded the patients who (1) had open fractures, (2) had medical conditions that delayed the surgery for fracture for more than 2 weeks after trauma, (3) had missing data or loss of follow-up 1-year post-trauma, and (4) had haemoglobinopathies such as thalassemia. The outcome was the incidence of surgical site infection (SSI) during 1 year postoperative. SSI was defined as (1) superficial SSI if any one of the following criteria were met: only infiltrates the local skin or subcutaneous tissue or involves an inpatient who needed antibiotic treatment for wound problems, including redness, swelling, heat, or pain, regardless of any microbiology results; (2) deep SSI if any one of the following criteria were met: deep fascia infection, local site abscess needing surgical debridement, persistent effusion or dehiscence of the wound, or the necessity of the implant removal or revision for infection control. These patients’ wound exudates were sent for causative agent culture and sensitivity [22].

### Statistical Analysis

SPSS for Windows (version 23.0; IBM, Armonk, NY, USA) was used for statistical analyses. Categorical variables are presented in terms of the frequency and proportion, and continuous variables are presented in terms of the mean ± standard deviation. A Chi-square test and an independent *t*-test were used to compare the men’s and women’s demographic characteristics. Multiple logistic regression was performed to determine the independent predictors for postoperative 1-year SSI. We used these models to identify the correlation of preoperative and postoperative HbA1c/AC glucose levels with postoperative SSI. A receiver-operating characteristic (ROC) curve was plotted to assess the diagnostic performance of preoperative and postoperative (3-month) HbA1c and AC glucose in SSI prediction. Sensitivity, specificity, and the area under the curve (AUC) were calculated, and we determined the optimal thresholds that indicate SSI, recurring to the Youden index. A two-sided *p* value < 0.05 indicated statistical significance.

## 3. Results

The patients’ demographic characteristics are shown in Table 1.

Specifically, the sample comprised 604 patients with more women than men (64%), a mean age of 71.21, a mean BMI of 25.26, a mean preop HbA1c of 7.55, and a mean preop blood glucose level of 146.83. In total, 50 of them (8.3%) were newly diagnosed T2DM. The comorbidities included hypertension (82%), dyslipidemia (44.2%), CAD (31.5%), cerebrovascular accident (27%), hepatic disease (18.4%), CRF (31%), and RA (39.2%). Among the 604 patients, 84 (13.9%) developed SSI. There were 62 superficial SSI and 22 deep SSI. The range of the occurrence of superficial SSI from the date of surgery was from 2 to 74 days, while that of deep SSI was from 11 to 110 days. The most frequent cultured pathogens were Staphylococcus aureus and Pseudomonas aeruginosa, as shown in Table 2.

Four adjusted models were constructed to evaluate the independent effect of the four glycemic indicators on SSI. Models 1 to 4 included preoperative HbA1c, postoperative HbA1c, preoperative blood glucose, and postoperative blood glucose, respectively. Our results are summarized in Table 3.

All four glycemic indicators were associated with increased rates of SSI in their adjusted models, with a preoperative HbA1c odds ratio (OR) of 1.77 (95% CI: 1.52–2.02; *p* < 0.001), postoperative HbA1c OR of 1.43 (95% CI: 1.28–1.59; *p* < 0.001), preoperative blood glucose OR of 1.02 (95% CI: 1.01–1.02; *p* < 0.001), and postoperative blood glucose of 1.01 (95% CI: 1.01–1.02; *p* < 0.001). Hypertension was associated with decreased rates of SSI in model 1 (OR, 0.43; 95% CI: 0.21–0.86; *p* = 0.017) and model 2 (OR, 0.44; 95% CI: 0.23–0.84; *p* = 0.013). Age was associated with decreased rates of SSI in model 3 (OR, 0.97; 95% CI: 0.95–0.99; *p* = 0.021) and model 4 (OR, 0.98; 95% CI: 0.95–0.99; *p* = 0.026). In model 3, CAD was associated with increased rates of SSI (OR, 2.25; 95% CI: 1.02–4.95; *p* = 0.045), whereas cerebrovascular accident was associated with decreased rates (OR, 0.27; 95% CI: 0.10–0.77; *p* = 0.014). Notably, comorbidity with RA was associated with increased rates of SSI in all four models, with an OR of 2.5 (95% CI: 1.39–4.48 *p* = 0.002) in model 1, an OR of 2.53 (95% CI: 1.48–4.33 *p* = 0.001) in model 2, an OR of 2.97 (95% CI: 1.48–5.99 *p* = 0.002) in model 3, and an OR of 2.85 (95% CI:1.55–5.23 *p* = 0.001) in model 4.

ROC curves were generated to compute the area under the ROC (AUROC), the best threshold, and the corresponding sensitivity and specificity for all four glycemic indicators on SSI. The AUROC results indicated that preoperative HbA1c (AUROC = 0.793; 95% CI: 0.730–0.856) (Figure 1A), preoperative blood glucose (AUROC = 0.773; 95% CI: 0.694–0.851) (Figure 1B), and postoperative blood glucose (AUROC = 0.709; 95% CI: 0.629–0.790) were highly predictive (AUROC > 0.7) of SSI (Figure 1C), whereas postoperative HbA1c (Figure 1D) (AUROC = 0.648; 95% CI: 0.570–0.726) was moderately predictive. With regard to the HbA1c threshold, a preoperative HbA1c of less than 7.850 predicted the absence of SSI with 78.6% sensitivity and 73.6% specificity (Figure 1A), and a postoperative HbA1c threshold of less than 6.650 did so with 73.8% sensitivity and 43.3% specificity (Figure 1D). For blood glucose, a preoperative blood glucose level of less than 130.50 predicted the absence of SSI with 76.3% sensitivity and 58.9% specificity (Figure 1B), and a postoperative blood glucose threshold of less than 148.5 did so with 65.2% sensitivity and 72.1% specificity (Figure 1C).

## 4. Discussion

This study indicated that increased perioperative blood glucose and HbA1c are associated with higher SSI rates. We also provided thresholds for each glycemic index, with high to moderate predictability confirmed by AUROC. Additionally, we showed that patients are at greater risk of SSI if previously diagnosed with RA. Consistent with similar research [4,7,16,19], preoperative HbA1c and blood glucose are associated with a higher risk of SSI. In contrast to the many studies that have focused on preoperative glycemic management and its association with surgical outcomes, the present study is the first to demonstrate that higher postoperative HbA1c is associated with increased rates of SSI in patients with T2DM with orthopedic trauma. The result indicated that other than preoperative glycemic status, which is commonly focused on, postoperative glycemic status is also a key factor affecting SSI. Our findings are significant for T2DM orthopedic patients in traumatic settings since our results imply that they may benefit from glycemic management even postoperatively. The Pearson’s correlation coefficient between pre-HbA1c and post-HbA1c was 0.934, whereas the correlation coefficient between pre-glucose and post-glucose was 0.605. Medium to high correlation was observed. All of these indices were associated with SSI. In order to determine which index was the most appropriate to be included in the model, we adopted forward logistic regression and we found the first covariate selected was pre-HbA1c. Thus, we suggested pre-HbA1c was the best value for the prediction of SSI as per the statistical model 1 in Table 3 in this study.

Blankush et al. (2016) demonstrated that a preoperative HbA1c level may not be independently associated with the risk of postoperative infection in 2200 patients undergoing nonemergent procedures, but it may be useful for predicting an increased risk of infectious complications for subgroup analysis [23]. Werner et al. (2019) analyzed data from 7958 patients with diabetes who underwent open surgery for carpal tunnel syndrome and revealed that increased HbA1c levels are associated with increased SSI rates among these patients with a perioperative HbA1c level of between 7 and 8 mg/dL—which serves as a threshold indicating SSI [24]. These findings may suggest that the control of DM for elective surgery should be monitored not only during the preoperative period but also during the postoperative period. The values for monitoring blood glycemic control should include parameters other than HbA1c, which indicates the quality of long-term blood glucose control; these other indicators include AC glucose or glycated albumin [25], which may indicate the quality of short-term blood glucose control. For the patients who need emergent surgery for fracture fixation, no relevant guidelines for DM control exist in the prevention of postoperative SSI. Considering the variation among the patients and their traumatic mechanisms, we focused our study group on patients older than 50 who received surgery for unilateral lower limb fractures to achieve a more than satisfactory prediction quality for perioperative HbA1c and AC glucose. A preoperative HbA1c of less than 7.850%, with postoperative HbA1c less than 6.650%, preoperative blood glucose less than 130.50 mg/dL, and postoperative blood glucose of less than 148.5 mg/dL indicated the importance of DM control quality postoperatively in the prevention of SSI for these patients with trauma. Our data on the perioperative glycemic control were similar to the suggested glycemic control data of studies focusing on elective lower limb surgery for total joint replacement [26,27]. Wexler et al. conducted a study in 2008 of 695 patients admitted to an acute care hospital that revealed a prevalence of unrecognized probable diabetes of 18% based on HbA1c levels higher than 6.1%, with 5% of them having HbA1c over 6.5% [28]. In this study, we have 8.3% newly diagnosed T2DM, and we gave prompt medication for diabetes control before the fracture surgery. This study also revealed that better perioperative blood glucose level control for good HbA1c and blood glucose level at postoperative three months is just as crucial as preoperative data of HbA1c and blood glucose level for the prevention of SSI.

Additionally, our study showed that, among all comorbidities, RA is the only factor significantly associated with SSI in all models; that may be due to the disease itself, associated commodities, or the use of immunosuppressive medication [29]. Our findings imply that healthcare professionals must consider SSI when approaching patients with T2DM comorbid with RA, which is becoming increasingly vital since, in this aging society, we expect an increase in the number of patients comorbid with both T2DM and RA. Hypertension and age seemed to be significantly associated with SSI in some models in our results. Previous studies have mentioned that hypertension, peripheral vascular disease, and advanced age may have a negative impact on increasing SSI in ankle fractures [30,31]. Another study revealed that hypertension, as one of the crucial diseases in metabolic syndrome, may be associated with increased odds of complications but decreased odds of in-hospital mortality in patients with hip fractures [32]. Older patients may have more comorbidities related to SSI, more skin complications, and poorer immune protection from bacteria than younger patients. We also believe that hypertension may also have the most adequate threshold for decreasing perioperative complications, including SSI. The adequate control of the patient’s underlying diseases, such as RA, hypertension, and peripheral vascular disease, especially in older adults, is just as crucial as the control of HbA1c and blood glucose level in T2DM to decrease the incidence of SSI.

This study had several limitations. First, the number of cases in our 10-year period was relatively small because many emergent-surgery patients did not have adequate monitoring data on HbA1C or AC glucose for follow-up. Second, this study did not include the patients’ DM medication, lifestyle, or dietary habits, which may influence their blood glycemic control quality. Third, we did not do further subgroup analysis for the different surgical methods for different kinds of lower limb fractures. Despite these limitations, this study provides the prediction thresholds that have good discrimination quality for preoperative and postoperative (3-month) HbA1c and AC glucose; these thresholds can be used as a reference for perioperative DM control and infection control precaution in preventing postoperative SSI among middle-aged and older adult patients. This is the first study focusing on perioperative blood glycemic monitoring for emergent lower limb fracture surgery. Further long-term follow-up on the influence of the control of T2DM parameters on the incidence of postoperative complications of limb fractures among large cohort groups will be our future research.

## 5. Conclusions

We demonstrated that higher preoperative and postoperative blood glucose and HbA1c and RA are associated with a higher risk of SSI for T2DM orthopedic trauma patients. The threshold for predicting SSI were (1) preoperative HbA1c > 7.850%; (2) postoperative HbA1c > 6.650%; (3) preoperative blood glucose > 130.50 mg/dL; and (4) postoperative blood glucose > 148.5 mg/dL. These glycemic index thresholds obtained from ROC have high to moderate predictive performance for postoperative 1-year SSI, and preoperative HbA1c seemed to be the best predictive value for SSI.

## Figures and Tables

**Figure 1 jcm-11-05552-f001:**
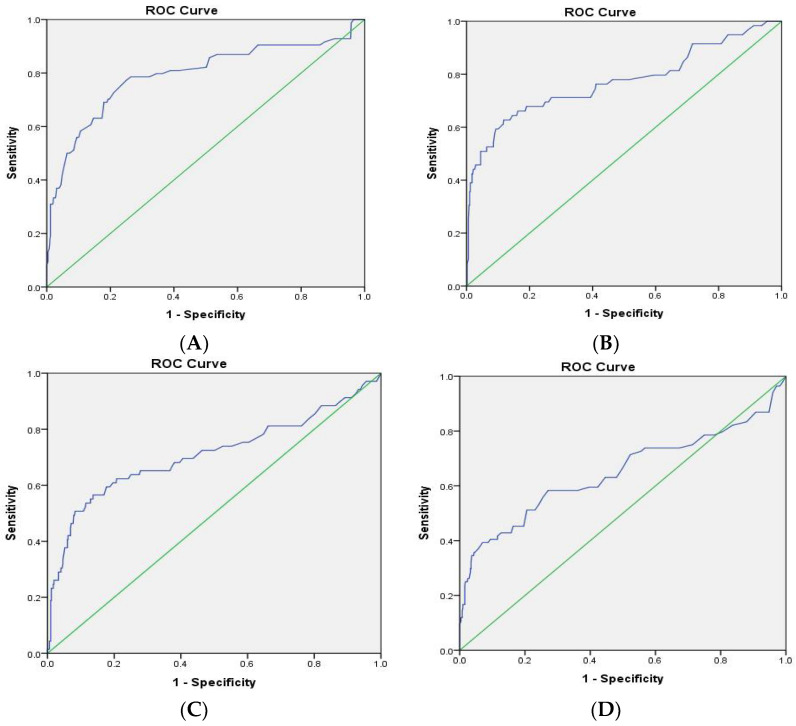
Receiver operating characteristic (ROC) curves were generated to compute the area under the ROC (AUROC), the best threshold, and the corresponding sensitivity and specificity for all four glycemic indicators on the incidence of surgical site infection during postoperative 1 year: The green lines represents the reference lines and the blue lines represented the differentiation power at the different thresholds of the glycemic indicators.(**A**) preoperative HbA1c (AUROC = 0.793; 95% CI: 0.730–0.856); (**B**) preoperative blood glucose (AUROC = 0.773; 95% CI: 0.694–0.851); (**C**) postoperative blood glucose (AUROC = 0.709; 95% CI: 0.629–0.790); (**D**) postoperative HbA1c (AUROC = 0.648; 95% CI: 0.570–0.726).

**Table 1 jcm-11-05552-t001:** Demographics of the T2DM patients (*n* = 604).

	Male	Female	Total	*p* Value
N	215	389	604	
Age	66.98 ± 15.08	73.55 ± 11.87	71.21 ± 13.46	<0.001 *
BMI	25.10 ± 4.50	25.34 ± 4.78	25.26 ± 4.68	0.551
Preoperative HbA1c (%)	7.60 ± 2.24	7.53 ± 1.99	7.55 ± 2.08	0.680
Preoperative AC Glucose (mg/dL)	150.26 ± 61.08	144.96 ± 54.05	146.83 ± 56.64	0.284
Medical comorbidities				
HTN (%)	167 (77.7%)	328 (84.3%)	495 (82.0%)	0.042 *
Dyslipidemia (%)	88 (40.9%)	179 (46.0%)	267 (44.2%)	0.228
CAD (%)	70 (32.6%)	120 (30.8%)	190 (31.5%)	0.665
CVA (%)	54 (25.1%)	109 (28.0%)	163 (27.0%)	0.441
Hepatic disease (%)	51 (23.7%)	60 (15.4%)	111 (18.4%)	0.012 *
CRF (%)	71 (33.0%)	116 (29.8%)	187 (31.0%)	0.415
RA (%)	78 (36.3%)	159 (40.9%)	237 (39.2%)	0.268
Surgical site infection (%)	36 (16.7%)	48 (12.3%)	84 (13.9%)	0.134

T2DM = Type 2 diabetes, BMI = body mass index, HbA1c = glycated hemoglobin, AC = Ante Cibum (before meals), HTN = Hypertension, CAD = coronary artery disease, CVA = cerebrovascular accident, CRF = Chronic renal failure, RA = rheumatoid arthritis, Data are presented as n or mean ± standard deviation, * *p* value < 0.05 was considered statistically significant after test.

**Table 2 jcm-11-05552-t002:** Characteristics of postoperative surgical site infection (*n* = 84).

	Superficial	Deep
N	62	22
Days after ORIF	2–74	11–110
Pathogens	methicillin-resistant *Staphylococcus aureus* (6)	methicillin-resistant *Staphylococcus aureus* (4)
	methicillin-susceptible *Staphylococcus aureus* (34)	methicillin-susceptible *Staphylococcus aureus* (9)
	*Pseudomonas aeruginosa* (12)	*Pseudomonas aeruginosa* (3)
	*Enterococcus faecalis* (3)	*Enterobacter cloacae* (1)
	Mixed bacteria (4)	*Escherichia coli* (1)
		*Acinetobacter baumannii* (1)
		Mixed bacteria (3)

ORIF: open reduction and internal fixation for fractures.

**Table 3 jcm-11-05552-t003:** Factors associated with postoperative surgical site infection.

	Crude		Adjusted (Model 1)		Adjusted (Model 2)		Adjusted (Model 3)		Adjusted (Model 4)	
	OR (95% CI)	*p* Value	OR (95% CI)	*p* Value	OR (95% CI)	*p* Value	OR (95% CI)	*p* Value	OR (95% CI)	*p* Value
Age	0.97 (0.95, 0.99)	<0.001 *	0.99 (0.97, 1.02)	0.635	0.99 (0.97, 1.01)	0.222	0.97 (0.95, 0.99)	0.021 *	0.98 (0.95, 0.99)	0.026 *
Sex (M vs. F)	1.43 (0.89, 2.28)	0.135	1.4 (0.78, 2.52)	0.264	1.26 (0.74, 2.17)	0.393	1.20 (0.59, 2.44)	0.611	1.12 (0.60, 2.07)	0.720
BMI	1.01 (0.96, 1.06)	0.711	1 (0.95, 1.07)	0.873	1.00 (0.95, 1.06)	0.950	0.98 (0.91, 1.05)	0.558	0.99 (0.93, 1.05)	0.729
Preoperative HbA1c	1.76 (1.56, 1.98)	<0.001 *	1.77 (1.56, 2.02)	<0.001 *	-	-	-	-	-	-
Postoperative HbA1c	1.42 (1.29, 1.57)	<0.001 *	-	-	1.43 (1.28, 1.59)	<0.001 *	-	-	-	-
Preoperative Glucose	1.02 (1.01, 1.02)	<0.001 *	-	-	-	-	1.02 (1.01, 1.02)	<0.001 *	-	-
Postoperative Glucose	1.01 (1.01, 1.02)	<0.001 *	-	-	-	-	-	-	1.01 (1.01, 1.02)	<0.001 *
HTN vs. None	0.49 (0.29, 0.83)	0.008 *	0.43 (0.21, 0.86)	0.017 *	0.44 (0.23, 0.84)	0.013 *	0.53 (0.23, 1.24)	0.146	0.51 (0.24, 1.08)	0.077
Dyslipidemia vs. None	1.31 (0.83, 2.08)	0.250	0.96 (0.52, 1.77)	0.900	1.08 (0.63, 1.88)	0.774	0.80 (0.39, 1.63)	0.537	1.21 (0.66, 2.22)	0.543
CAD vs. None	1.25 (0.77, 2.03)	0.366	1.73 (0.91, 3.31)	0.097	1.67 (0.92, 3.03)	0.090	2.25 (1.02, 4.95)	0.045 *	1.44 (0.73, 2.87)	0.297
CVA vs. None	0.60 (0.34, 1.06)	0.080	0.58 (0.28, 1.21)	0.146	0.65 (0.33, 1.25)	0.192	0.27 (0.10, 0.77)	0.014 *	0.54 (0.25, 1.13)	0.102
Hepatic disease vs. None	1.15 (0.65, 2.05)	0.635	1.08 (0.54, 2.16)	0.821	0.93 (0.49, 1.79)	0.838	0.88 (0.37, 2.07)	0.767	0.96 (0.47, 1.95)	0.903
CRF vs. None	1.07 (0.65, 1.75)	0.801	1.60 (0.83, 3.05)	0.158	1.48 (0.82, 2.69)	0.196	1.26 (0.59, 2.71)	0.555	1.45 (0.75, 2.80)	0.275
RA vs. None	2.21 (1.38, 3.52)	0.001 *	2.50 (1.39, 4.48)	0.002 *	2.53 (1.48, 4.33)	0.001 *	2.97 (1.48, 5.99)	0.002 *	2.85 (1.55, 5.23)	0.001 *

OR = odd ratio, BMI = body mass index, HbA1c = glycated hemoglobin, HTN = Hypertension, CAD = coronary artery disease, CVA = cerebrovascular accident, CRF = Chronic renal failure, RA = rheumatoid arthritis, Data are presented as Odds ratio (95% CI). * *p* value < 0.05 was considered statistically significant after test.

## Data Availability

Data is contained within the article.

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
