# Peer review of "Correlation between the Control of Blood Glucose Level and HbA1C and the Incidence of Surgical Site Infection after Emergent Surgery for the Lower Limb Fracture among Type II DM Patients Aged More Than 50 Years Old"

_jcm, 2022, doi:10.3390/jcm11195552_

Round 1
Reviewer 1 Report
General comments
Authors tried to demonstrate that perioperative (pre and 3-months-post) glycemic index (blood glucose and HbA1c) are associated with surgical outcomes.
In addition, RA is associated with a higher risk of SSI for T2DM patients.
Authors demonstrated that this study may have great contribution for postoperative glycemic evaluation and management of patients with T2DM receiving elective orthopedic surgery; however, the authors are missing important data that readers need, as well as some discussion of the results obtained.
Specific comments
1) This study indicated that increased perioperative blood glucose and HbA1c are associated with higher SSI rates. However, the details of SSI (classification, time of occurrence, etc.) are not described. Author should be clearly indicated and presented in the Table.
2) This study compared the demographic characteristics for men and women in the Table 1. What is the significance of comparing men and women in this study?
3) The authors need to comment on why they focused on patients over 50 years of age.
4) In the discussion of important factors associated with SSI, the authors only discussed RA among all comorbidities. However, comments are also needed on factors associated with SSI, such as hypertension and age.
5) In this study, the AUROC study showed that both pre and post were associated with postoperative SSI, but would it be possible to assess that pre is more strongly associated with postoperative SSI than postoperative? Consideration is needed for both pre and post having relevance to SSI.
Author Response
1) This study indicated that increased perioperative blood glucose and HbA1c are associated with higher SSI rates. However, the details of SSI (classification, time of occurrence, etc.) are not described. The author should be clearly indicated and presented in the Table.
Ans: Thank you for your reminding. We have made a new table 2 for the characteristics of SSI cases and added the related description in the Result section:" The range of the occurrence of superficial SSI from date of surgery was from 2 to 74 days, while that of deep SSI was from 11 to 110 days. The most frequent cultured pathogens were Staphylococcus aureus and Pseudomonas aeruginosa (Table 2)."
|
Table 2. Characteristics of postoperative surgical site infection (n = 84). |
||
|
Superficial |
Deep |
|
|
N |
62 |
22 |
|
Days after ORIF |
2-74 |
11-110 |
|
Pathogens |
methicillin-resistant Staphylococcus aureus (6) |
methicillin-resistant Staphylococcus aureus (4) |
|
|
methicillin-susceptible Staphylococcus aureus (34) |
methicillin-susceptible Staphylococcus aureus (9) |
|
|
Pseudomonas aeruginosa (12) |
Pseudomonas aeruginosa (3) |
|
|
Enterococcus faecalis (3) |
Enterobacter cloacae (1) |
|
|
Mixed bacteria (4) |
Escherichia coli (1) |
|
|
|
Acinetobacter baumannii (1) |
|
|
|
Mixed bacteria (3) |
2) This study compared the demographic characteristics for men and women in the Table 1. What is the significance of comparing men and women in this study?
Ans: Thank you for your asking. Though the gender difference seemed not to be a significant risk factor for the incidence of SSI. We believe that gender was the basic pre-existing individual difference factor and it seemed to be adequate to be used to make comparative grouping for the presentation of the demographic data of this study participants.
3) The authors need to comment on why they focused on patients over 50 years of age.
Ans: Thank you for your asking. We considered this study to be focused on middle-aged and elderly adults because the impact and the prevalence of T2DM and their influence on surgical site infection seemed to be larger than those among the young patient group.
4) In the discussion of important factors associated with SSI, the authors only discussed RA among all comorbidities. However, comments are also needed on factors associated with SSI, such as hypertension and age.
Ans: Thank you for your suggestion. We originally did not discuss these two factors because they did not show statistical significance in all four analytical models. We have added the discussion below:" Hypertension and age seemed to be significantly associated with SSI in some models in our results. Previous studies have mentioned hypertension, peripheral vascular disease, and advanced age may have a negative impact on increasing SSI at the ankle fractures [29-30]. Another study revealed that hypertension, as one of the crucial diseases in metabolic syndrome, may be associated with increased odds of complications but decreased odds of in-hospital mortality in patients with hip fractures [31]. Older patients may have more comorbidities related to SSI, more skin complications, and poorer immune protection from the bacteria than younger patients. We also believe that hypertension may also have the most adequate threshold for decreasing perioperative complications, including SSI. The adequate control of the patient's underlying diseases, such as RA, hypertension, and peripheral vascular disease, especially in old adults, is just as crucial as the control of HbA1c and blood glucose level in T2DM to decrease the incidence of SSI. "
29. Sun Y, Wang H, Tang Y, Zhao H, Qin S, Xu L, Xia Z, Zhang F. Incidence and risk factors for surgical site infection after open reduction and internal fixation of ankle fracture: A retrospective multicenter study. Medicine (Baltimore). 2018 Feb;97(7):e9901. doi: 10.1097/MD.0000000000009901
30. Molina CS, Stinner DJ, Fras AR, Evans JM. Risk factors of deep infection in operatively treated pilon fractures (AO/OTA: 43). J Orthop. 2015 Feb 21;12(Suppl 1):S7-S13. doi: 10.1016/j.jor.2015.01.026.
31. Cichos KH, Churchill JL, Phillips SG, Watson SL, McGwin G Jr, Ghanem ES, Ponce BA. Metabolic syndrome and hip fracture: Epidemiology and perioperative outcomes. Injury. 2018 Nov;49(11):2036-2041. doi: 10.1016/j.injury.2018.09.012.
5) In this study, the AUROC study showed that both pre and post were associated with postoperative SSI, but would it be possible to assess that pre is more strongly associated with postoperative SSI than postoperative? Consideration is needed for both pre and post having relevance to SSI.
Ans: Thank you for your reminding. Indeed, the Pearson's correlation coefficient between pre-HbA1c and post-HbA1c was 0.934, whereas the correlation coefficient between pre-glucose and post-glucose was 0.605. Medium to high correlation was observed. All of these indices were associated with SSI. In order to determine which index was the most appropriate to be included in the model, we adopted forward logistic regression and we found the first covariate selected was pre-HbA1c. Thus, we suggested pre-HbA1c was the most suitable to be included in the model (model 1 in table 3). We have added this point into Discussion and Conclusion.
Reviewer 2 Report
This is a very interesting article.
It would be interesting how much the risk of SSI increased with the different thresholds of HbA1c.
Authors should also discuss about the value of HbA1c in trauma patients who has not been diagnosized of diabetes or new diabetes diagnoses.
Author Response
Thank you very much for your suggestions. We have added that" 50 of them (8.3%) were newly-diagnosed T2DM." in the Result section. We also added the below sentences into the Discussion section:" Wexler et al. conducted a study in 2008 revealed a prevalence of unrecognized probable diabetes of 18% based on HbA1c levels higher than 6.1%, with 5% of them having HbA1c over 6.5% in 695 patients admitted to the acute care hospital[27]. In this study, we have 8.3% newly-diagnosed T2DM, and we gave prompt medication for diabetes control before the fracture surgery. This study also revealed that better perioperative blood glucose level control for good HbA1c and blood glucose level at postoperative three months is just as crucial as preoperative data of HbA1c and blood glucose level for the prevention of SSI."